# Characterization of Seventeen Complete Mitochondrial Genomes: Structural Features and Phylogenetic Implications of the Lepidopteran Insects

**DOI:** 10.3390/insects13110998

**Published:** 2022-10-31

**Authors:** Meiling Cheng, Yi Liu, Xiaofeng Zheng, Rusong Zhang, Kaize Feng, Bisong Yue, Chao Du, Chuang Zhou

**Affiliations:** 1Key Laboratory of Bioresources and Eco-Environment (Ministry of Education), College of Life Sciences, Sichuan University, Chengdu 610064, China; 2State Forestry and Grassland Administration Key Laboratory of Conservation Biology for Rare Animals of the Giant Panda State Park, China Conservation and Research Center for the Giant Panda, Dujiangyan 611830, China; 3Key Laboratory of Sichuan Province for Fishes Conservation and Utilization in the Upper Reaches of the Yangtze River, Neijiang Normal University, Neijiang 641000, China; 4Baotou Teachers College, Baotou 014060, China

**Keywords:** Lepidoptera, moth, complete mitochondrial genome, structural feature, phylogenetic analysis

## Abstract

**Simple Summary:**

The moth is a large group with high species richness in Lepidoptera, and many species of them play important roles in human’s life. However, there remains limited understanding for the phylogenetic relationship of moths due to absent taxon samples and genetic sequences. In this study, we described seventeen new species involving five major lineages of moths and discussed the phylogeny based on the mitochondrial genome data. The results revealed a special mitogenome characteristic in *Sphragifera sigillata* that possessed two control regions located between l-rRNA and trnV and between s-rRNA and trnM, respectively. Besides, the phylogeny presented a different relationship from previous studies. These findings will broaden our knowledge of moths and make contributions to genetics, biology, and ecology.

**Abstract:**

Lepidoptera (moths and butterflies) are widely distributed in the world, but high-level phylogeny in Lepidoptera remains uncertain. More mitochondrial genome (mitogenome) data can help to conduct comprehensive analysis and construct a robust phylogenetic tree. Here, we sequenced and annotated 17 complete moth mitogenomes and made comparative analysis with other moths. The gene order of trnM-trnI-trnQ in 17 moths was different from trnI-trnQ-trnM of ancestral insects. The number, type, and order of genes were consistent with reported moths. The length of newly sequenced complete mitogenomes ranged from 14,231 bp of *Rhagastis albomarginatus* to 15,756 bp of *Numenes albofascia*. These moth mitogenomes were typically with high A+T contents varied from 76.0% to 81.7% and exhibited negative GC skews. Among 13 protein coding genes (PCGs), some unusual initiations and terminations were found in part of newly sequenced moth mitogenomes. Three conserved gene-overlapping regions and one conserved intergenic region were detected among 17 mitogenomes. The phylogenetic relationship of major superfamilies in Macroheterocera was as follows: (Bombycoidea + Lasiocampoidea) + ((Drepanoidea + Geometroidea) + Noctuoidea)), which was different from previous studies. Moreover, the topology of Noctuoidea as (Notodontidae + (Erebidae + Noctuidae)) was supported by high Bayesian posterior probabilities (BPP = 1.0) and bootstrapping values (BSV = 100). This study greatly enriched the mitogenome database of moth and strengthened the high-level phylogenetic relationships of Lepidoptera.

## 1. Introduction

Lepidoptera is one of the widely recognizable insect orders in the world, widely distributed in all terrestrial habitats ranging from desert to rainforest and from lowland grasslands to mountain plateaus [1]. There are about 180,000 Lepidoptera species described, including moths and butterflies [2,3], wherein moths make up the vast majority (approximate 160,000) of this order [4]. Many moths are closely related to human life; for example, the domesticated silkworm moth is an important source of silk, and thus, they are viewed as valuable economic resource [5]. For some species of the Pyralidae, their larvae are used for the biological control of weeds in place of herbicides because they feed on a single species or limited range of plants [6]. Additionally, many species in Tortricidae, Noctuidae, and Pyralidae are major pests in agriculture. For example, the larvae of the genus *Spodoptera* (armyworms) and *Helicoverpa* (corn earworm) in Noctuidae can cause extensive damage to certain crops [7]. Although a considerable number of moths has been studied, many moths have not been described and investigated yet.

Lepidoptera contains 43 superfamilies, of which only one superfamily belongs to butterflies, and all the rest belong to moths [8]. Of them, Macroheterocera is one of the high-level phylogenetic clades within moths that have been defined [8]. Macroheterocera includes six superfamilies (Mimallonoidea, Drepanoidea, Geometroidea, Noctuoidea, Bombycoidea, and Lasiocampoidea) [8]. However, the phylogeny of the superfamily Macroheterocera clade is unstable; for instance, the Phylogenetic status of Mimallonoidea and Drepanoidea are highly controversial. Most studies place Mimallonoidea as basal to other macroheterocerans, followed by Drepanoidea [9,10,11]. Nevertheless, another study [12] uncovered that Drepanoidea is more closely related to ((Bombycoidea + Lasiocampoidea) + Noctuoidea). In addition, the phylogenetic relationship among other macroheterocerans remains confusing. The relationship of (Noctuoidea + (Geometroidea + (Bombycoidea + Lasiocampoidea))) was supported by most studies based on various data, such as mitogenome sequences [13,14], multi-gene sequences [15], and 741 genes from transcriptome sequences [9]. However, Kawahara and Breinholt [16] suggested a different phylogeny, i.e., ((Noctuoidea + Geometroidea) + (Bombycoidea + Lasiocampoidea)), according to transcriptome sequences from 2696 genes and mitogenome sequences, respectively. It seems that the use of various molecular markers and limited data lead to these phylogenetic differences in Macroheterocera.

The application of molecular markers in phylogenetics significantly accelerates the process to understand the evolution of life [17,18]. Whole mitogenomes contain high phylogenetic information, which provides increased resolution at different taxonomic levels [8]. Many studies have used mitogenomes to address complex phylogenetic of moths, such as Geometroidea [19], Noctuoidea [20], Macroheterocera [11], and major families in moths [21]. Thus, mitogenomes are potential molecular markers for phylogenetic analysis. Although the mitogenomes of about 120 macroheteroceran species or subspecies have been published [10,11,12,13,19,21], only a small part of mitogenomes have been described in comparative analyses, and there is still a need for newer mitogenomes to be described. Since more mitogenome data can help to perform comprehensive analysis, we have obtained the complete mitogenomes of 17 moth species that belonged to six superfamilies (one species of Cossoidea, one of Lasiocampoidea, five of Bombycoidea, one of Drepanoidea, three of Geometroidea, and six of Noctuoidea). These newly sequenced mitogenomes were used for phylogenetic inference of several superfamilies and families of Macroheterocera clade through Bayesian inference (BI) and maximum likelihood (ML) method. Our study would further enrich the moth mitogenome database and strengthen the phylogenetic relationships among major superfamilies and families in Macroheterocera.

## 2. Materials and Methods

### 2.1. Taxon Sampling

The adult specimens of seventeen Lepidoptera species were collected in Mount Qingcheng (Dujiangyan city, Sichuan Province, China; 103°34′ E, 30°54′ N). The collected specimens were quickly preserved in absolute ethanol and then stored at −20 °C prior for DNA extraction. We first performed morphological identification of these samples by traditional methods. Then, for more precise species identification of the samples, we performed blast searches of the nucleotide collection (nr/nt) database of the NCBI based on cytochrome c oxidase I (COI) mitochondrial gene. We acquired the best-fit and targeted mitochondrial sequences by BLAST searches under at least 98%.

### 2.2. Mitogenome Sequencing, Assembly, and Annotation 

The total genomic DNA was extracted from muscle using the Genomic DNA Extraction Kit (Sangon Biotech, Shanghai, China), and the quality of total DNA was checked with 1% agarose gels. Extracted DNA was sequenced using the next-generation sequencing technology. The whole-genome shotgun method was used to get paired-end libraries and sequence on an Illumina MiSeq platform (Novogene, Beijing, China), with the target insert size of 500 bp. 

The assembly and annotation of mitogenome data were done by Mitoz v.2.3 [22]. Firstly, the raw fastq files were filtered by the filter module in Mitoz v.2.3 [22] to obtain clean data. Then, based on the obtained clean data, the assembly and annotation module in Mitoz v.2.3 [22] was employed for assembly and annotation of the complete mitogenomes with genetic code 5 and clade Arthropoda and other parameters as default. Finally, gene boundaries were further confirmed and aligned against the published mitogenome sequences of moths using MEGA v.7.0 [23].

The complete mitogenomes of 17 moth species were submitted to GenBank, and the accession numbers in bold beginning with OP are shown in Table 1.

### 2.3. Comparative Mitogenome Analyses

The exact location of the control region was determined by confirming the boundary of neighboring genes. The relative synonymous codon usage (RSCU), base composition, and amino acid composition of the newly sequenced moth mitogenomes were analyzed using MEGA v. 7.0 [23] and Phylosuite v.1.2.2. [24]. The number of non-synonymous substitutions per nonsynonymous site (Ka) and synonymous substitutions per synonymous site (Ks) for each PCG were calculated using DNASP 6 [25], with exclusion of stop codons and codons with alignment gaps. The skew of the nucleotide composition was calculated with the formulas: AT-skew = (A − T) / (A + T) and GC-skew = (G − C)/(G + C) [26].

### 2.4. Phylogenetic Analysis

Utilizing the PCG sequences of mitogenomes to infer phylogenetic relationships is an informative taxonomic strategy [10]. Thus, we used the PCG dataset of 94 moth mitogenomes (17 newly sequenced mitogenomes and 77 reported mitogenomes) to infer phylogenetic tree with two Trichoptera species as the outgroup (Table 1). DAMBE v.7.3.5 [27] was used to test nucleotide substitution saturation and plot transition and transversion substitutions against the TN93 distance for the data set before inferring the phylogenetic trees. 

The plugins in Phylosuite v.1.2.2. [24] were used to extract the 13 PCGs data for the phylogenetic construction. The alignment of PCGs was conducted using MAFFT [28], according to a codon-based model, and gaps and ambiguous sites were then removed by Gblocks [29]. All alignments were then concatenated into a single data matrix using the concatenate sequence function in Phylosuite. The best-fit partitioning strategy and models were selected using the greedy search by PartitionFinder [30]. As a result, twelve subsets partition schemes for the PCG data matrix were obtained for BI analyses (subset1-8,10,12: GTR + I + R, atp6-8, cox1-3, nad1, nad2, nad4, and nad6; subset9,11: TIM + I + G, nad4l, and nad5). The MrBayes v3.2.7 [31] was used to phylogenetic inference using BI methods with four independent Markov chains run for 10,000,000 generations and below an average standard deviation value of 0.01. The IQ-TREE [32] in Phylosuite was employed to phylogenetic inference using ML methods with automatic model prediction and 1000 ultrafast bootstraps. 

## 3. Results

### 3.1. Mitogenome Structure and Organization

The length of newly sequenced 17 complete mitogenomes ranged from 14,231 bp in *Rhagastis albomarginatus* to 15,756 bp in *Numenes albofascia*. All newly sequenced mitogenomes were circular and double-stranded molecules and possessed 37 genes (22 tRNAs, 13 PCGs, 2 rRNAs (l-rRNA + s-rRNA)) and the A + T-rich region (the control region (CR)). There were 23 genes (including 14 tRNAs and 9 PCGs) located in the majority strand (J-strand) and 14 genes (including 8 tRNAs, 4 PCGs and 2 rRNAs) located in the minority strand (N-strand) (Appendix A). Only one CR was located between s-rRNA and trnM in 16 mitogenomes except *Sphragifera sigillata*, with two CRs located between l-rRNA and trnV and between s-rRNA and trnM, respectively (Figure 1). The gene order and orientation of the 17 mitogenomes was consistent with that of other moth mitogenomes [21] in possessing the same gene order (trnM-trnI-trnQ) and differed from that of ancestral insects (trnI-trnQ-trnM) [33] (Figure 1). 

The nucleotide composition of these 17 mitogenomes was highly towards A and T nucleotides. The A/T content ranged from 76.0% in *Gazalina chrysolopha* to 81.7% in *Phyllosphingia dissimilis* (Table 2). The GC skews were negative in 16 mitogenomes except for positive GC skews in *Lassaba albidaria*, and the absolute value of GC skew was greater than 0.1, while the AT skew below 0.1. (Table 2). 

### 3.2. Protein-Coding Genes

In total, thirteen PCGs were detected, of which nine PCGs (nad2, cox1, cox2, atp8, atp6, cox3, nad3, nad6, and cytb) were located in J-strand and four PCGs (nad5, nad4, nad4l, and nad1) in N-strand. The nucleotide composition of PCGs in both J-strand and N-strand both exhibited A + T bias. The A/T content of PCGs in J-strand varied from 72.2% in *G. chrysolopha* to 81.3% in *L. albidaria* (Appendix A). The A/T content of PCGs in N-strand was between 75.8% in *G. chrysolopha* and 82.4% in *P. dissimilis* (Appendix A). The AT skews and GC skews of J-strand PCGs were negative in 16 mitogenomes except for negative AT skew and positive GC skews in *L. albidaria* (Appendix A). The AT skew and GC skews of N-strand PCGs were negative and positive (respectively) in 16 mitogenomes except for positive AT skew and negative GC skews in *L. albidaria* (Appendix A).

In 17 mitogenomes, PCGs mainly started with ATN (ATA, ATT, ATG, and ATC) and ended with TAA. However, some unusual initiations and terminations were found. nad4 of *Zeuzera pyrina* and nad1 of *G. chrysolopha*, *Dolbina paraexacta,* and *Dolbina inexacta* started with GTG. nad1 of *Menophra* sp. and *Psyra falcipennis* started with TTG. cox1 of several species started with CGA, TTG, AAA, and AAG. Several PCGs (nad2, atp6, nad3, nad5, nad4l, and nad1) of some species ended with TAG. cox2, nad4, and nad5 of some species ended with incomplete termination codons T or TA (Table 3). 

The analysis result of RSCU appeared that there were codon usage preferences on UUA, UUU, UAU, AAA, AAU, AUA, and AUU (Figure 2). The most frequent coding amino acids and the predilection codons were leu2 (UAA) and ile (AUU) (Appendix A). Both RSCU and most frequent coding amino acids showed A + T predilection. To investigate evolutionary patterns of PCGs, the values of Ka, Ks, and Ka/Ks were calculated. The PCGs of 94 moth insects were employed to calculate the Ka, Ks, and the ratio of Ka/Ks. The Ka/Ks value for atp8 was the highest, followed by nad5, nad4l, nad6, nad1, nad2, nad4, nad3, atp6, cytb, cox3, cox2, and cox1 (Figure 3).

### 3.3. Gene Overlapping Regions and Intergenic Regions

Among seventeen moth mitogenomes, we detected three conserved gene-overlapping regions with the nucleotide lengths of 5 bp, 7 bp, and 8 bp, respectively. The 5 bp conserved gene-overlapping region was “TCTAA”, located at cox1 and trnL2 junction. The 7 bp overlap of “ATGATAA” was searched between atp6 and atp8. The 8 bp overlap between trnC and trnW was composed of the “AAGCCTTA”. In addition, the conserved intergenic region of “ATACTAA” was recognized between nad1 and trnS2 among 17 mitogenomes. The length of the intergenic region located between nad2 and trnQ ranged from 48 bp to 87 bp in 17 newly sequenced moth mitogenomes, which showed high A/T contents (Figure 4).

### 3.4. Phylogenetic Analyses

We used the concatenated nucleotide sequences of the 13 PCGs to construct phylogenetic relationships and tested saturation. The result showed that the transition and transversion were linearly associated with the corrected genetic distance (Figure 5), indicating that the data set was not saturated and qualified for phylogenetic trees construction.

We analyzed the phylogenetic relationships of the six superfamilies in order Lepidoptera, with two Trichoptera species as outgroup. The phylogenetic trees based on ML and BI methods generated almost identical topology in family and superfamily level. The relationship among superfamilies was (Cossoidea + (Gelechioidea + (Bombycoidea + Lasiocampoidea) + ((Drepanoidea + Geometroidea) + Noctuoidea)))), and the relationship among families was (Cossoidae + (Autostichidae + ((Lasiocampidae + Sphingidae) + ((Drepanidae + Geometridae) + (Notodontidae + (Noctuidae + Erebidae)))))) (Figure 6 and Figure 7). The monophyly of each superfamily and family were strongly supported by Bayesian posterior probabilities (BPP = 1.00) and moderately supported by bootstrapping values (BSV > 73). The phylogenetic relationships of several major superfamilies in Macroheterocera clade were (Bombycoidea + Lasiocampoidea) + ((Drepanoidea + Geometroidea) + Noctuoidea)). Lasiocampoidea and Bombycoidea formed a sister relationship with high support value (BPP = 1.00, BSV = 91). (Drepanoidea + Geometroidea) + Noctuoidea was strongly supported (BPP = 1.00). 

Phylogenetic relationships of newly sequenced 17 species were supported with general high value of BPP = 1.00, with BSV > 66. *Z. pyrina*, belonging to family Cossidae, formed a sister group with *Zeuzera multistrigata*. *Paralebeda femorata* was a member of family Lasiocampidae. *R. albomarginatus*, *P. dissimilis*, *H. circumflexa*, *D. paraexacta,* and *D. inexacta* were part of family Sphingidae, in which *R. albomarginatus* and *Theretra japonica*, *P. dissimilis*, *Rhodoprasina callantha* and *H. circumflexa*, *D. paraexacta,* and *D. inexacta* separately formed into sister relationships. *D. hyalina* was assigned to the family Drepanidae. *P. falcipennis*, *Menophra* sp., and *L. albidaria* were in family Geometridae, and the sister relationship of *P. falcipennis* and *Menophra* sp. obtained high support values, but the position of *L. albidaria* in Geometridae was different in ML and BI trees. *Zaranga tukuringra* and *G. chrysolopha* belonged to family Notodontidae, and the sister relationship of *G. chrysolopha* and *Thaumetopoea pityocampa* was supported with high values. *S. sigillata* and *Olivenebula oberthueri* were located in family Noctuidae. *N. albofascia* and *Asota tortuosa* were assigned to family Erebidae, and *A. tortuosa* and *Asota plana lacteata* formed a sister group. 

## 4. Discussion

### 4.1. Mitogenome Organization, Gene Overlaps, and Spacers

The length of 17 newly sequenced, complete mitogenomes varied from 14,231 bp to 15,756 bp, which was mainly due to differences in intergenic spacers. Except that the order of trnM-trnI-trnQ in 17 moths was different from trnI-trnQ-trnM of ancestral insects [33], the number, type, and order of genes were consistent with reported moths [11,21]. The rearrangement of trnQ was typical in moths [34,35,36], which could be resulted from the tandem duplication–random loss (TDRL) [37]. Interestingly, the latest research found for the first time in chironomids that mitochondrial gene rearranged from trnI-trnQ-trnM to trnI-trnM-trnQ [38]. The base composition distribution varied from 76.0% to 81.7%, with a high A + T content being typical in insect mitogenomes [39]. Negative GC skew in newly sequenced mitogenome was also found in other moths [11,21].

We performed a comparative analysis among 17 mitogenomes, and we identified three conserved gene-overlapping regions and one conserved intergenic region. Among them, the 7 bp gene-overlapping region “ATGATAA” was between atp6 and atp8 and was a character commonly found in Lepidoptera [20,40] as well as other insects [41,42]. The gene-overlapping regions “TCTAA” and “AAGCCTTA” that were separately located at cox1 and trnL2 and trnC and trnW were also detected in Lepidoptera: Papilionoidea [40]. Except for the junction of trnS2 and nad1, i.e., TTACTAA, of *Menophra* sp., which was also recognized in Diptera: Drosophila [43], the intergenic region at trnS2 and nad1 junction in the other 16 species was “ATACTAA”. The intergenic of “ATACTAA” had been widely reported in insect mitogenomes [20,43,44] as being responsible for mitochondrion transcription [20,43,45]. Further attention is needed regarding the intergenic region of trnS2 and nad1 in *P. femorata* and *P. dissimilis*, with part of this intergenic junction being at the 3′ end of the nad1. Furthermore, the spacer sequence located between the trnQ and nad2 genes was extremely A + T-rich [46]. This region was widely present in Lepidoptera and may even be regarded as a synapomorphy of lepidopteran species [20,47].

### 4.2. Protein-Coding Genes

PCGs mainly started with ATN and ended with TAA, which was typical of insect mitogenomes [11,21]. The start codon of cox1 varied greatly and was found to be unstable in various arthropod mitogenomes [11,21,34,48]. nad1 and nad4 of some species started with GTG, which was also found in other animal mitogenomes [49]. nad1 of *Menophra* sp. and *P. falcipennis* started with TTG, which also existed in other moths [44]. nad2, atp6, nad3, nad5, nd4l, and nad1 of some species ended with TAG, which was also found in other moths [44,50]. cox2, nad4, and nad5 of some species ended with incomplete termination codons T and TA, which occurred in other moths [21,44,50,51]. The incomplete stop codons were thought to be modified into complete TAA through posttranscriptional polyadenylation in the period of the mRNA maturation [52].

The ratio of Ka/Ks is a typical indicator of evolutionary rate [53,54]. The Ka/Ks value for atp8 was the highest, and cox1 was the lowest. This indicated that the atp8 was under the least selection pressure and cox1 under the highest selection pressure among the moth mitochondrial PCGs. All Ka/Ks of 13 PCGs were below 1, indicating these genes were evolving under purifying selection and were suitable for investigating phylogenetic relationships [11].

### 4.3. Phylogenetic Relationships among the Major Macroheterocera Groups

The backbone phylogeny of the Macroheterocera clade in Lepidoptera remained precarious. In our study, based on the 13 PCGs data set of mitogenomes, the phylogenetic relationships of several major superfamilies in Macroheterocera clade were (Bombycoidea + Lasiocampoidea) + ((Drepanoidea + Geometroidea) + Noctuoidea)), which showed a difference from previous reports. Especially in the phylogenetic status of Drepanoidea, despite limited taxon sampling, our result supported that Drepanoidea was more related to Geometroidea through high BPP (=1.0). Most previous reports placed Drepanoidea in basic phylogenetic position, followed by Mimallonoidea in the Macroheterocera clade [9,10,11]. However, Heikkilä et al. [12] supported that Drepanoidea was more closely related to (Bombycoidea + Lasiocampoidea) + Noctuoidea. The limited date of Drepanoidea should be the most important reason for these deviations, and adding data from each taxon at the same time to construct a comprehensive and robust phylogenetic tree will help resolve these biases. In addition, regarding the phylogenetic relationship between Noctuoidea, Geometroidea, Bombycoidea, and Lasiocampoidea also seemed to be unstable. Some research supported the relationship as Noctuoidea + (Geometroidea+ (Bombycoidea + Lasiocampoidea)) [9,13,14,15]. Others recovered the following relationships: (Noctuoidea + Geometroidea) + (Bombycoidea + Lasiocampoidea) [11,16]. Our findings were more supportive of (Noctuoidea + Geometroidea) + (Bombycoidea + Lasiocampoidea). 

The phylogeny position of Geometroidea and Lasiocampoidea in Macroheterocera was labile. Some studies [43] favored (Geometroidea + (Noctuoidea + Bombycoidea)) as forming the Macrolepidoptera “core”. However, another study [46] supported that Geometroidea was closely related to Bombycoidea and then Noctuoidea in ML and BI analysis. In contrast, Kawahara and Breinholt [16] and Yang et al. [11] supported Geometroidea being closer to Noctuoidea. In this study, based on ML and BI analysis, Geometroidea was closely related to Drepanoidea and then Noctuoidea, with high BPP (=1.0), but only one species was added in Drepanoidea to infer the phylogeny, and more data were needed to confirm this systematic relationship. In addition, the phylogenetic relationships of Lasiocampidea and Bombycoidea remained confused. Some studies treated Lasiocampoidea as a family of Bombycoidea [55,56,57]. The other studies supported the sister relationship of Lasiocampoidea and Bombycoidea [9,11,16]. In this study, Lasiocampoidea and Bombycoidea separately clustered one branch with high support value, and it was more likely to favor sister groups of Lasiocampoidea and Bombycoidea.

Noctuoidea (∼42,000 species) is by far the largest lepidopteran superfamily [8], with long-presented, difficult phylogenetic problems [58]. Zahiri et al. [59], based on seven mitochondrial genes and one nuclear gene, gave the relationship of families in Noctuoidea as (Notodontidae + (Euteliidae + (Noctuidae +(Erebidae + Nolidae)))). Evidence from the complete mitogenomes favored the hypothesis of (Notodontidae + (Erebidae + (Nolidae + (Euteliidae + Noctuidae)))) [13]. Then, the result of Regier et al. [58] well-supported the phylogenetic relationship of (Notodontidae + (Erebidae + (Noctuidae + (Euteliidae + Nolidae)))), which was analyzed with 5–19 genes (6.7–18.6 kb) in 74 noctuoids. Based on nucleotide sequence and amino acid sequence of 13 PCGs of mitogenomes, the family-level topology of the phylogenetic analyses was described as (Notodontidae + (Erebidae + (Nolidae + (Euteliidae + Noctuidae)))) [35]. Inconsistent molecular markers, limited sample sizes, or different analytical methods might cause the different phylogenetic results [35]. Regardless, based on the above results, Notodontidae as the basal position of Noctuoidea is relatively stable, and families in the superfamily Noctuoidea remain monophyletic. In our analyses, the ML and BI both strongly supported (BPP = 1.00, BSV = 100) the monophyly of Notodontidae, Erebidae, and Noctuidae and the topology of phylogenetic relationship as (Notodontidae + (Erebidae + Noctuidae)), which was partially consistent with the result of Zhu et al. [35]. 

In conclusion, based on the 13 PCGs data set of mitogenomes, our analyses supported the topology of the phylogenetic relationship as (Notodontidae + (Erebidae + Noctuidae)) with high support value. Whole mitogenomes contain high phylogenetic information, which are proven to be potential molecular markers for moths’ high-level phylogenetic analysis, while it is worth noting the poor signal for phylogenetic reconstruction at the subfamily level in the chironomids [60]. 

## 5. Conclusions

We sequenced and annotated 17 complete moth mitogenomes, and we performed comparative analysis with other moths. In total, 37 mitogenomes genes (13 PCGs + 22 tRNAs + 2 rRNAs) and the CR were detected. Except that the order of trnM-trnI-trnQ in 17 moths was different from trnI-trnQ-trnM of ancestral insects, the number, type, and order of genes were consistent with reported moths. The length of newly sequenced complete mitogenomes ranged from 14,231bp of *Rhagastis albomarginatus* to 15,756bp of *Numenes albofascia*. These moth mitogenomes were typically with high A + T content ranging from 76% to 81.7% and exhibited negative GC skews. Among 13 PCGs, some unusual initiations (AAA, AAG, CGA, TTG, and GTG) and terminations (TA and T) were found in some newly sequenced moths. The atp8 was under the least selection pressure, while cox1 was under the highest selection pressure among the moth mitochondrial PCGs. Three conserved gene-overlapping regions and one conserved intergenic region were detected among 17 mitogenomes. The topology relationship of major superfamilies in Macroheterocera was as follows: (Bombycoidea + Lasiocampoidea) + ((Drepanoidea + Geometroidea) + Noctuoidea)), which differed from those relationships published previously. Our analyses supported the monophyly of Notodontidae, Erebidae, and Noctuidae and the topology of the phylogenetic relationship as (Notodontidae + (Erebidae + Noctuidae)) with high support value. However, there are still many problems in the higher-level phylogeny of Macroheterocera remaining unsolved, and further adding more data from each taxon to construct a robust phylogenetic tree is still needed. In short, this study greatly enriched the mitogenome database of moths and again strengthened the high level of phylogenetic relationships of Lepidoptera. 

## Figures and Tables

**Figure 1 insects-13-00998-f001:**
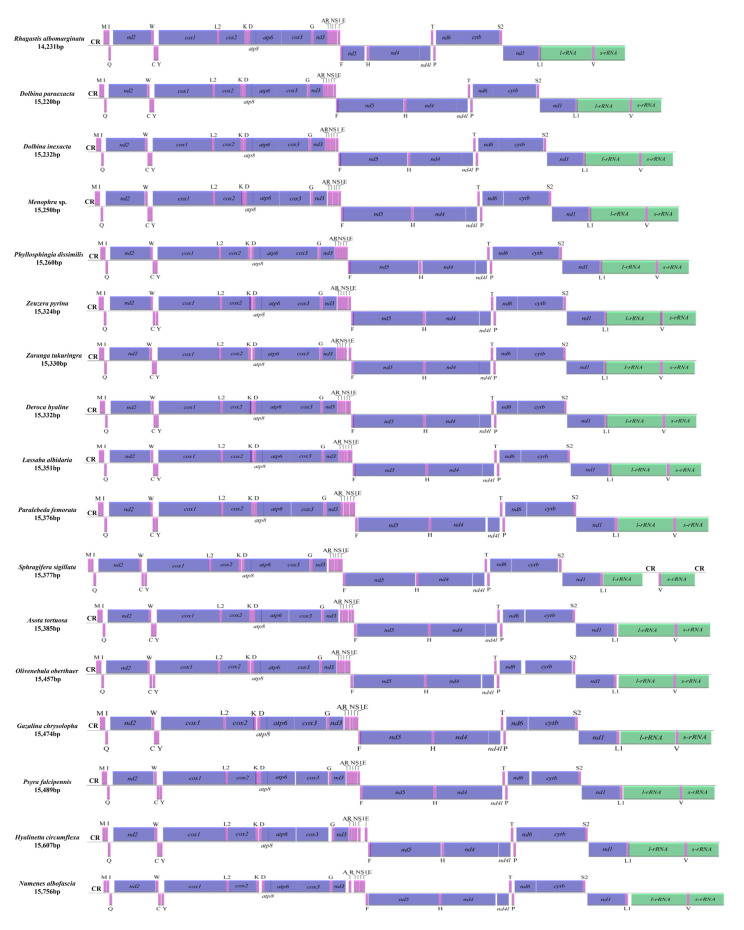
Linear map of the complete mitogenomes of 17 moths. CR, putative control region. Note: The nad5 in the mitogenome of *R. albomarginatus* may need further validation.

**Figure 2 insects-13-00998-f002:**
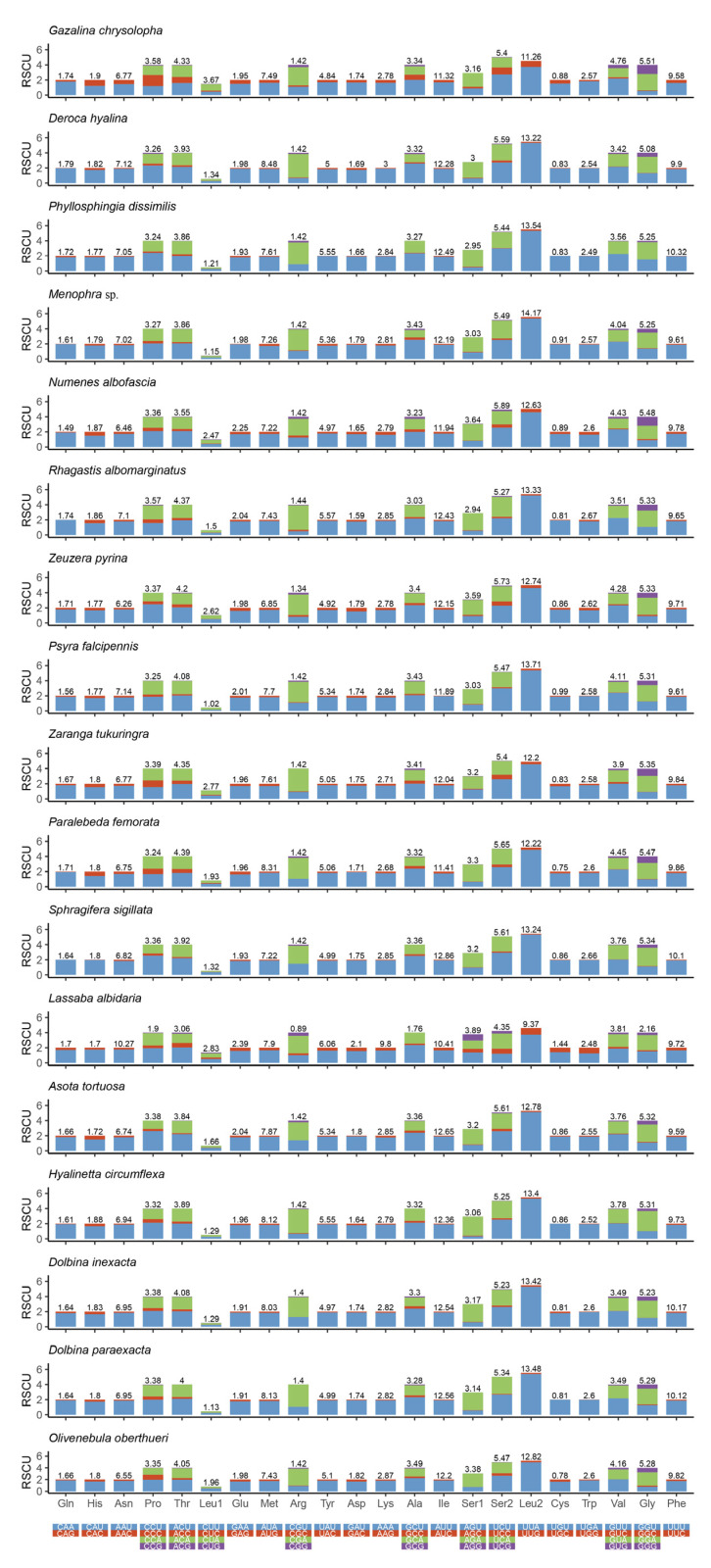
Relative synonymous codon usage (RSCU) and codon distribution in PCGs of 17 moths mitogenomes. Codon families are indicated below the *x*-axis.

**Figure 3 insects-13-00998-f003:**
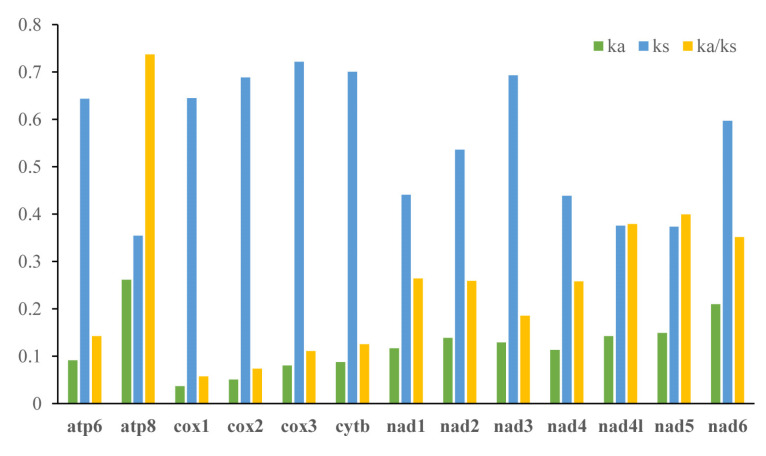
Evolutionary rates of 13 PCGs in 94 moths. Rate of non-synonymous substitutions (Ka), rate of synonymous substitutions (Ks), and ratio of rate of non-synonymous substitutions to rate of synonymous substitutions (Ka/Ks) are calculated for each PCG.

**Figure 4 insects-13-00998-f004:**
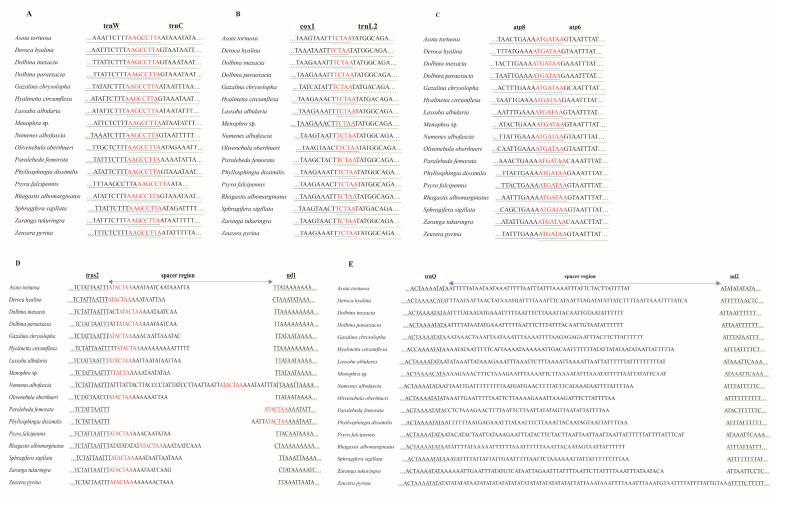
The overlapping region and spacers in 17 new mitogenomes. The underline color of genes is consistent with that of their corresponding sequences. (**A**–**C**) represent the overlapping region between trnW and trnC, cox1 and trnL2, and atp8 and atp6, respectively. The nucleotides marked red indicate the sequence of overlapping region. (**D**). The intergenic spacer between trnS2 and nd1. The nucleotides marked red indicate the conserved motif sequence. (**E**). The intergenic spacer between trnQ and nd2.

**Figure 5 insects-13-00998-f005:**
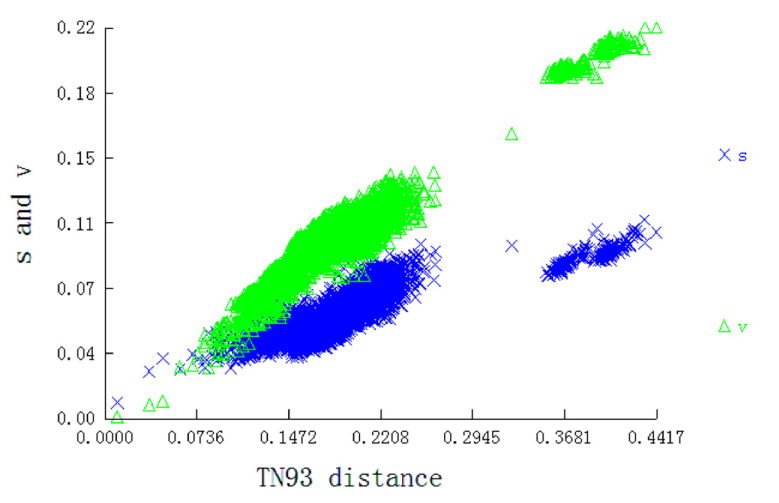
Substitution patterns of 13 PCGs matrices. The graph represents the increase in TN93 distance.

**Figure 6 insects-13-00998-f006:**
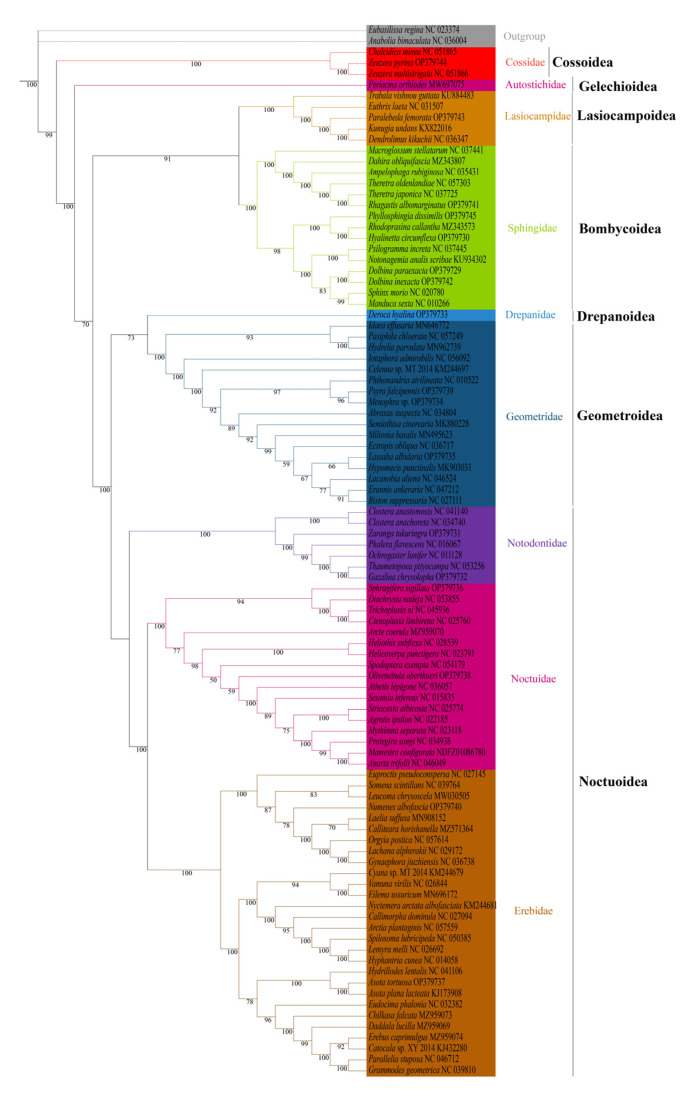
Phylogenetic tree inferred from DNA sequences of 13 PCGs using maximum likelihood (ML) analysis. Numbers on branches are bootstrap percentages.

**Figure 7 insects-13-00998-f007:**
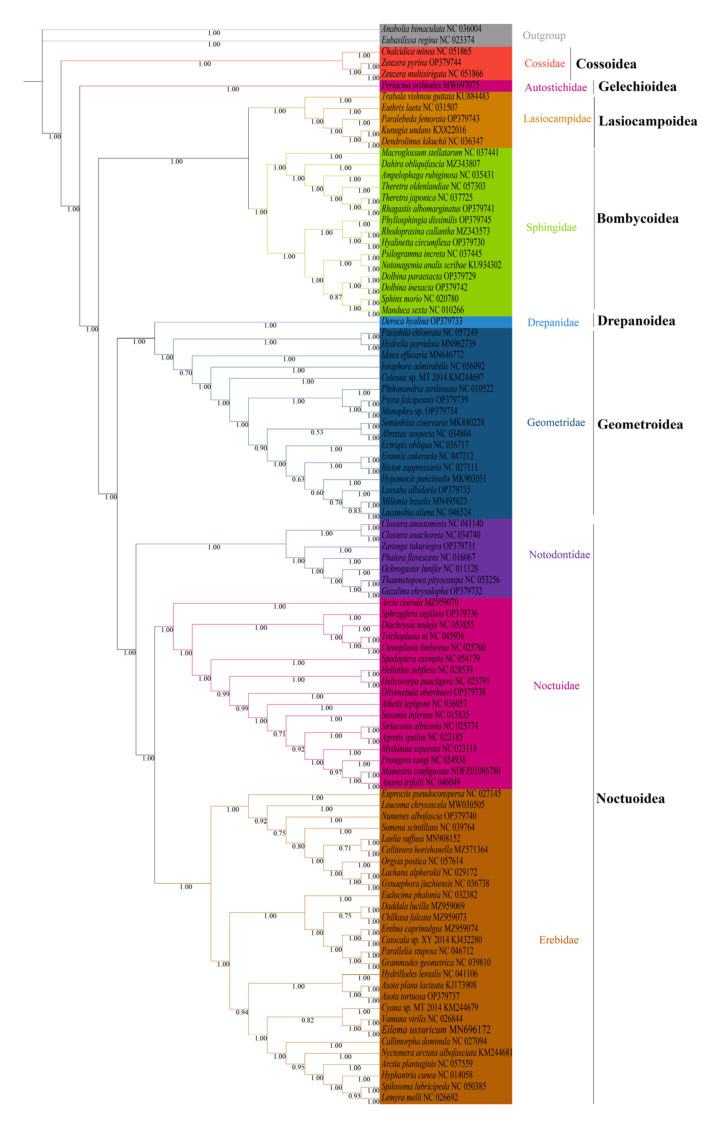
Phylogenetic tree inferred from DNA sequences of 13 PCGs using Bayesian inference (BI) analysis. The number on the branches indicates Bayesian posterior probabilities.

**Table 1 insects-13-00998-t001:** Species information of mitochondrial phylogenetic analysis.

	Order	Superfamily	Family	Species	Accession ID
Ingroup	Lepidoptera	Drepanoidea	Drepanidae	*Deroca hyalina*	OP379733
	Lepidoptera	Noctuoidea	Erebidae	*Erebus caprimulgus*	MZ959074
	Lepidoptera	Noctuoidea	Erebidae	*Lemyra melli*	NC_026692
	Lepidoptera	Noctuoidea	Erebidae	*Orgyia postica*	NC_057614
	Lepidoptera	Noctuoidea	Erebidae	*Somena scintillans*	NC_039764
	Lepidoptera	Noctuoidea	Erebidae	*Spilosoma lubricipeda*	NC_050385
	Lepidoptera	Noctuoidea	Erebidae	*Vamuna virilis*	NC_026844
	Lepidoptera	Noctuoidea	Erebidae	*Callimorpha dominula*	NC_027094
	Lepidoptera	Noctuoidea	Erebidae	*Calliteara horishanella*	MZ571364
	Lepidoptera	Noctuoidea	Erebidae	*Catocala* sp. XY 2014	KJ432280
	Lepidoptera	Noctuoidea	Erebidae	*Chilkasa falcata*	MZ959073
	Lepidoptera	Noctuoidea	Erebidae	*Cyana* sp. MT 2014	KM244679
	Lepidoptera	Noctuoidea	Erebidae	*Laelia suffusa*	MN908152
	Lepidoptera	Noctuoidea	Erebidae	*Leucoma chrysoscela*	MW030505
	Lepidoptera	Noctuoidea	Erebidae	*Parallelia stuposa*	NC_046712
	Lepidoptera	Noctuoidea	Noctuidae	*Spodoptera exempta*	NC_054179
	Lepidoptera	Noctuoidea	Noctuidae	*Striacosta albicosta*	NC_025774
	Lepidoptera	Noctuoidea	Noctuidae	*Diachrysia nadeja*	NC_053855
	Lepidoptera	Noctuoidea	Noctuidae	*Heliothis subflexa*	NC_028539
	Lepidoptera	Noctuoidea	Noctuidae	*Protegira songi*	NC_034938
	Lepidoptera	Noctuoidea	Notodontidae	*Phalera flavescens*	NC_016067
	Lepidoptera	Noctuoidea	Notodontidae	*Thaumetopoea pityocampa*	NC_053256
	Lepidoptera	Noctuoidea	Notodontidae	*Ochrogaster lunifer*	NC_011128
	Lepidoptera	Noctuoidea	Notodontidae	*Zaranga tukuringra*	OP379731
	Lepidoptera	Noctuoidea	Notodontidae	*Gazalina chrysolopha*	OP379732
	Lepidoptera	Noctuoidea	Notodontidae	*Clostera anachoreta*	NC_034740
	Lepidoptera	Noctuoidea	Notodontidae	*Clostera anastomosis*	NC_041140
	Lepidoptera	Bombycoidea	Sphingidae	*Macroglossum stellatarum*	NC_037441
	Lepidoptera	Bombycoidea	Sphingidae	*Manduca sexta*	NC_010266
	Lepidoptera	Bombycoidea	Sphingidae	*Notonagemia analis scribae*	KU934302
	Lepidoptera	Bombycoidea	Sphingidae	*Psilogramma increta*	NC_037445
	Lepidoptera	Bombycoidea	Sphingidae	*Rhodoprasina callantha*	MZ343573
	Lepidoptera	Bombycoidea	Sphingidae	*Sphinx morio*	NC_020780
	Lepidoptera	Bombycoidea	Sphingidae	*Theretra japonica*	NC_037725
	Lepidoptera	Bombycoidea	Sphingidae	*Theretra oldenlandiae*	NC_057303
	Lepidoptera	Bombycoidea	Sphingidae	*Dolbina paraexacta*	OP379729
	Lepidoptera	Bombycoidea	Sphingidae	*Rhagastis albomarginatus*	OP379741
	Lepidoptera	Bombycoidea	Sphingidae	*Dolbina inexacta*	OP379742
	Lepidoptera	Bombycoidea	Sphingidae	*Phyllosphingia dissimilis*	OP379745
	Lepidoptera	Bombycoidea	Sphingidae	*Ampelophaga rubiginosa*	NC_035431
	Lepidoptera	Bombycoidea	Sphingidae	*Dahira obliquifascia*	MZ343807
	Lepidoptera	Bombycoidea	Sphingidae	*Hyalinetta circumflexa*	OP379730
	Lepidoptera	Cossoidea	Cossidae	*Zeuzera multistrigata*	NC_051866
	Lepidoptera	Cossoidea	Cossidae	*Zeuzera pyrina*	OP379744
	Lepidoptera	Cossoidea	Cossidae	*Chalcidica minea*	NC_051865
	Lepidoptera	Gelechioidea	Autostichidae	*Periacma orthiodes*	MW697075
	Lepidoptera	Geometroidea	Geometridae	*Menophra* sp.	OP379734
	Lepidoptera	Geometroidea	Geometridae	*Lassaba albidaria*	OP379735
	Lepidoptera	Geometroidea	Geometridae	*Ectropis obliqua*	NC_036717
	Lepidoptera	Geometroidea	Geometridae	*Erannis ankeraria*	NC_047212
	Lepidoptera	Geometroidea	Geometridae	*Hydrelia parvulata*	MN962739
	Lepidoptera	Geometroidea	Geometridae	*Hypomecis punctinalis*	MK903031
	Lepidoptera	Geometroidea	Geometridae	*Idaea effusaria*	MN646772
	Lepidoptera	Geometroidea	Geometridae	*Milionia basalis*	MN495623
	Lepidoptera	Geometroidea	Geometridae	*Pasiphila chloerata*	NC_057249
	Lepidoptera	Geometroidea	Geometridae	*Phthonandria atrilineata*	NC_010522
	Lepidoptera	Geometroidea	Geometridae	*Semiothisa cinerearia*	MK880228
	Lepidoptera	Geometroidea	Geometridae	*Psyra falcipennis*	OP379739
	Lepidoptera	Geometroidea	Geometridae	*Abraxas suspecta*	NC_034804
	Lepidoptera	Geometroidea	Geometridae	*Biston suppressaria*	NC_027111
	Lepidoptera	Geometroidea	Geometridae	*Celenna* sp. MT 2014	KM244697
	Lepidoptera	Geometroidea	Geometridae	*Iotaphora admirabilis*	NC_056092
	Lepidoptera	Geometroidea	Geometridae	*Lacanobia aliena*	NC_046524
	Lepidoptera	Lasiocampoidea	Lasiocampidae	*Dendrolimus kikuchii*	NC_036347
	Lepidoptera	Lasiocampoidea	Lasiocampidae	*Kunugia undans*	KX822016
	Lepidoptera	Lasiocampoidea	Lasiocampidae	*Trabala vishnou guttata*	KU884483
	Lepidoptera	Lasiocampoidea	Lasiocampidae	*Paralebeda femorata*	OP379743
	Lepidoptera	Lasiocampoidea	Lasiocampidae	*Euthrix laeta*	NC_031507
	Lepidoptera	Noctuoidea	Erebidae	*Arctia plantaginis*	NC_057559
	Lepidoptera	Noctuoidea	Erebidae	*Eudocima phalonia*	NC_032382
	Lepidoptera	Noctuoidea	Erebidae	*Grammodes geometrica*	NC_039810
	Lepidoptera	Noctuoidea	Erebidae	*Hydrillodes lentalis*	NC_041106
	Lepidoptera	Noctuoidea	Erebidae	*Nyctemera arctata albofasciata*	KM244681
	Lepidoptera	Noctuoidea	Erebidae	*Asota tortuosa*	OP379737
	Lepidoptera	Noctuoidea	Erebidae	*Numenes albofascia*	OP379740
	Lepidoptera	Noctuoidea	Erebidae	*Asota plana lacteata*	KJ173908
	Lepidoptera	Noctuoidea	Erebidae	*Daddala lucilla*	MZ959069
	Lepidoptera	Noctuoidea	Erebidae	*Eilema ussuricum*	MN696172
	Lepidoptera	Noctuoidea	Erebidae	*Euproctis pseudoconspersa*	NC_027145
	Lepidoptera	Noctuoidea	Erebidae	*Gynaephora jiuzhiensis*	NC_036738
	Lepidoptera	Noctuoidea	Erebidae	*Hyphantria cunea*	NC_014058
	Lepidoptera	Noctuoidea	Erebidae	*Lachana alpherakii*	NC_029172
	Lepidoptera	Noctuoidea	Noctuidae	*Agrotis ipsilon*	NC_022185
	Lepidoptera	Noctuoidea	Noctuidae	*Helicoverpa punctigera*	NC_023791
	Lepidoptera	Noctuoidea	Noctuidae	*Mamestra configurata*	NDFZ01086780
	Lepidoptera	Noctuoidea	Noctuidae	*Mythimna separata*	NC_023118
	Lepidoptera	Noctuoidea	Noctuidae	*Sesamia inferens*	NC_015835
	Lepidoptera	Noctuoidea	Noctuidae	*Trichoplusia ni*	NC_045936
	Lepidoptera	Noctuoidea	Noctuidae	*Sphragifera sigillata*	OP379736
	Lepidoptera	Noctuoidea	Noctuidae	*Olivenebula oberthueri*	OP379738
	Lepidoptera	Noctuoidea	Noctuidae	*Anarta trifolii*	NC_046049
	Lepidoptera	Noctuoidea	Noctuidae	*Arcte coerula*	MZ959070
	Lepidoptera	Noctuoidea	Noctuidae	*Athetis lepigone*	NC_036057
	Lepidoptera	Noctuoidea	Noctuidae	*Ctenoplusia limbirena*	NC_025760
Outgroup	Trichoptera		Limnephilidae	*Anabolia bimaculata*	NC_036004
	Trichoptera		Phryganeidae	*Eubasilissa regina*	NC_023374

**Table 2 insects-13-00998-t002:** The nucleotide composition of 17 mitogenomes.

Species	Size (bp)	T(U) (%)	C (%)	A (%)	G (%)	AT (%)	GC (%)	GT (%)	AT Skew	GC Skew
*Gazalina chrysolopha*	15,474	36.4	16.0	39.6	8.0	76.0	24.0	44.4	0.043	−0.335
*Deroca hyalina*	15,332	41.2	10.7	40.2	7.9	81.4	18.6	49.1	−0.012	−0.153
*Phyllosphingia dissimilis*	15,260	40.7	10.9	41.0	7.4	81.7	18.3	48.1	0.005	−0.191
*Menophra* sp.	15,250	39.9	11.3	41.0	7.8	80.9	19.1	47.7	0.013	−0.189
*Numenes albofascia*	15,756	38.6	13.5	40.3	7.6	78.9	21.1	46.2	0.021	−0.281
*Rhagastis albomarginatus*	14,231	40.5	11.7	40.3	7.5	80.8	19.2	48.0	−0.003	−0.217
*Zeuzera pyrina*	15,324	39.1	13.2	39.8	7.9	78.9	21.1	47.0	0.009	−0.256
*Psyra falcipennis*	15,489	39.3	11.1	41.8	7.8	81.1	18.9	47.1	0.031	−0.175
*Zaranga tukuringra*	15,330	37.5	13.5	41.2	7.8	78.7	21.3	45.3	0.047	−0.265
*Paralebeda femorata*	15,376	38.5	12.8	40.9	7.8	79.4	20.6	46.3	0.030	−0.247
*Sphragifera sigillata*	15,377	41.3	10.8	40.3	7.6	81.6	18.4	48.9	−0.013	−0.176
*Lassaba albidaria*	15,351	41.4	7.9	39.8	10.9	81.2	18.8	52.3	−0.020	0.165
*Asota tortuosa*	15,385	40.7	11.9	39.9	7.5	80.6	19.4	48.2	−0.011	−0.227
*Hyalinetta circumflexa*	15,607	41.0	11.1	40.6	7.3	81.6	18.4	48.3	−0.005	−0.211
*Dolbina inexacta*	15,232	39.6	11.5	41.3	7.6	80.9	19.1	47.2	0.022	−0.200
*Dolbina paraexacta*	15,220	39.8	11.2	41.4	7.6	81.2	18.8	47.4	0.020	−0.192
*Olivenebula oberthueri*	15,457	39.8	12.4	39.9	7.9	79.7	20.3	47.7	0.000	−0.220

**Table 3 insects-13-00998-t003:** The start and end codons of PCGs in 17 mitogenomes.

Species	ND2	COX1	COX2	ATP8	ATP6	COX3	ND3	ND5	ND4	ND4L	ND6	CYTB	ND1
*Zaranga tukuringra*	ATT/TAA	CGA/TAA	ATG/T	ATT/TAA	ATG/TAA	ATG/TAA	ATT/TAA	ATT/TAA	ATT/TA	ATG/TAA	ATA/TAA	ATG/TAA	ATG/TAG
*Gazalina chrysolopha*	ATT/TAG	CGA/TAA	ATG/TAA	ATC/TAA	ATG/TAA	ATG/TAA	ATT/TAA	ATT/TAA	ATG/TA	ATG/TAG	ATA/TAA	ATA/TAA	GTG/TAA
*Deroca hyalina*	ATT/TAA	TTG/TAA	ATA/TAA	ATC/TAA	ATG/TAA	ATG/TAA	ATC/TAA	ATA/TAA	ATG/TA	ATG/TAA	ATA/TAA	ATG/TAA	ATG/TAG
*Menophra* sp.	ATA/TAA	CGA/TAA	ATG/TAA	ATA/TAA	ATG/TAA	ATG/TAA	ATT/TAA	ATT/TAA	ATG/T	ATG/TAA	ATA/TAA	ATA/TAA	TTG/TAA
*Dolbina paraexacta*	ATT/TAA	AAA/TAA	ATG/T	ATC/TAA	ATG/TAA	ATG/TAA	ATT/TAA	ATT/TAA	ATG/TAA	ATG/TAA	ATG/TAA	ATG/TAA	GTG/TAA
*Lassaba albidaria*	ATA/TAA	ATT/TAA	ATG/TAA	ATC/TAA	ATG/TAA	ATG/TAA	ATA/TAA	ATT/TAA	ATG/TAA	ATG/TAA	ATA/TAA	ATG/TAA	ATA/TAA
*Sphragifera sigillata*	ATT/TAA	TTG/TAA	ATG/T	ATA/TAA	ATG/TAA	ATG/TAA	ATA/TAA	ATA/TAA	ATG/TA	ATG/TAA	ATT/TAA	ATA/TAA	ATG/TAA
*Asota tortuosa*	ATA/TAA	CGA/TAA	ATG/T	ATT/TAA	ATG/TAG	ATG/TAA	ATT/TAA	ATA/TAA	ATG/TA	ATG/TAA	ATT/TAA	ATG/TAA	ATG/TAA
*Olivenebula oberthueri*	ATT/TAA	CGA/TAA	ATG/T	ATA/TAA	ATG/TAA	ATG/TAA	ATA/TAA	ATT/TAA	ATG/TA	ATG/TAG	ATC/TAA	ATG/TAA	ATG/TAA
*Psyra falcipennis*	ATA/TAA	CGA/TAA	ATG/TAA	ATT/TAA	ATG/TAA	ATG/TAA	ATT/TAA	ATT/TA	ATG/T	ATG/TAA	ATA/TAA	ATG/TAA	TTG/TAA
*Numenes albofascia*	ATT/TAA	CGA/TAA	ATT/T	ATT/TAA	ATG/TAA	ATG/TAA	ATC/AAT	ATT/TAA	ATG/TA	ATG/TAA	ATA/TAA	ATA/TAA	ATA/TAA
*Rhagastis albomarginatus*	ATT/TAA	CGA/TAA	ATG/TA	ATC/TAA	ATG/TAA	ATG/TAA	ATT/TAG	ATT/TAA	ATG/TAA	ATG/TAA	ATG/TAA	ATG/TAA	ATG/TAG
*Dolbina inexacta*	ATT/TAA	AAA/TAA	ATG/T	ATC/TAA	ATG/TAA	ATG/TAA	ATT/TAA	ATT/TAA	ATG/TAA	ATG/TAA	ATG/TAA	ATG/TAA	GTG/TAA
*Paralebeda femorata*	ATA/TAA	AAG/TAA	ATA/T	ATC/TAA	ATG/TAA	ATG/TAA	ATT/TAA	ATT/TAG	ATG/TA	ATG/TAA	ATA/TAA	ATG/TAA	ATG/TAG
*Zeuzera pyrina*	ATT/TAA	CGA/TAA	ATG/TAA	ATT/TAA	ATG/TAA	ATG/TAA	ATT/TAA	ATT/TAG	GTG/TAA	ATG/TAA	ATC/TAA	ATG/TAA	ATG/TAA
*Phyllosphingia dissimilis*	ATT/TAA	ATT/TAA	ATG/T	ATT/TAA	ATG/TAA	ATG/TAA	ATC/TAA	ATT/TAG	ATG/TAA	ATG/TAA	ATG/TAA	ATG/TAA	ATG/TAG
*Hyalinetta circumflexa*	ATT/TAA	CGA/TAA	ATG/T	ATT/TAA	ATG/TAA	ATG/TAA	ATT/TAA	ATC/TAA	ATG/TAA	ATG/TAA	ATG/TAA	ATG/TAA	ATG/TAA

## Data Availability

The sequence data supporting the findings of this study is openly available in the project PRJNA892246 from the NCBI (https://www.ncbi.nlm.nih.gov/) accessed on 7 September 2022. The assembled mitogenomes accession numbers are listed in Table 1.

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
