# Peer review of "Characterization of Seventeen Complete Mitochondrial Genomes: Structural Features and Phylogenetic Implications of the Lepidopteran Insects"

_insects, 2022, doi:10.3390/insects13110998_

Round 1
Reviewer 1 Report
In their manuscript, Cheng et al. characterize seventeen complete mitochondrial genomes. They analyses structural features and discussed phylogenetic implications of 6 superfamilies if Lepidoptera (1 species of Cossoidea, 1 Lasiocampoidea, 5 Bombycoidea, 1Drepanoidea, 3 of Geometroidea, and 6 Noctuoidea). I think the study enriches the moth mitogenome database and thus provides important resources. However, in my opinion, the taxon sampling as well as the amount of data (only mitogenome, not nuclear genomic regions, only approx. 16,000 bp per species) is not enough to strengthen the phylogenetic relationships among major superfamilies and families in Macroheterocera and I wonder why the authors did not include all available Macroheterocera mitogenomes in their phylogenetic reconstructions. Authors state: "The relationship of (Noctuoidea + (Geometroidea + (Bombycoidea + Lasiocampoidea))) was supported by most studies based on various data such as mitogenome sequences, multi-gene sequences and 741 genes from transcriptome sequences. However, Kawahara and Breinholt suggested a different phylogeny as ((Noctuoidea + Geometroidea) + (Bombycoidea + Lasiocampoidea)) according to transcriptome sequences from 2,696 genes and mitogenome sequences, respectively. Thus, more genetic sequences are needed to construct more robust phylogenetic tree of Macroheterocerahard." So, I wonder if a few more mitogenome sequences are enough to further resolve the phylogeny. I suggest authors to reframe their paper in the sense that this is more a descriptive data note providing important data (mitogenomes) but not so much focusing much on the phylogenetic relationships -in my opinion-.
The methods are unclear, authors should clearly describe how they assembled and annotated the mitogenomes. (parameters of mitoz should be stated in the methods section). "We analyzed the phylogenetic relationships of the 6 superfamilies in order Lepidoptera, with two Trichoptera species as outgroup."
There needs to be a list with metadata for each specimen used in this study including, voucher, sampling coordinates, and species name. For example, it would be interesting which caddisfly mitogenomes were used.
Raw reads as well as mitogenomes need to be uploaded to NCBI SRA or a similar data portal, otherwise, the study is not reproducible and accession numbers need to be given in a table or appendix.
I could not access the figures and thus did not review these.
Author Response
Dear Reviewer,
Thank you for your comments concerning our manuscript entitled “Characterization of seventeen complete mitochondrial genomes: structural features and phylogenetic implications of the Lepidopteran insects ” (insects-1952476) . Those comments are valuable and very helpful. We have read through comments carefully and have made corrections. Based on the instructions provided in your letter, we uploaded the file of the revised manuscript. Revisions in the text are shown using red highlight for additions, and strikethrough font for deletions. The responses to the reviewer's comments are marked in red and presented following.
We would love to thank you for allowing us to resubmit a revised copy of the manuscript and we highly appreciate your time and consideration.
Sincerely.
- However, in my opinion, the taxon sampling as well as the amount of data (only mitogenome, not nuclear genomic regions, only approx. 16,000 bp per species) is not enough to strengthen the phylogenetic relationships among major superfamilies and families in Macroheterocera and I wonder why the authors did not include all available Macroheterocera mitogenomes in their phylogenetic reconstructions.
Response: We have obtained 16 complete mitogenomes of Macroheterocera, which are not distributed in all families of Macroheterocera. To infer the phylogeny of newly acquired species, we only added the available complete mitogenome samples of the family in which we have obtained new mitgenomes.
- Authors state: "The relationship of (Noctuoidea + (Geometroidea + (Bombycoidea + Lasiocampoidea))) was supported by most studies based on various data such as mitogenome sequences, multi-gene sequences and 741 genes from transcriptome sequences. However, Kawahara and Breinholt suggested a different phylogeny as ((Noctuoidea + Geometroidea) + (Bombycoidea + Lasiocampoidea)) according to transcriptome sequences from 2,696 genes and mitogenome sequences, respectively. Thus, more genetic sequences are needed to construct more robust phylogenetic tree of Macroheterocerahard." So, I wonder if a few more mitogenome sequences are enough to further resolve the phylogeny. I suggest authors to reframe their paper in the sense that this is more a descriptive data note providing important data (mitogenomes) but not so much focusing much on the phylogenetic relationships -in my opinion-.
Response: Thank you for your good suggestions. We have made some modifications in the second and third paragraphs of the Introduction. In fact, the most important meaning of our manuscript is to provide important mitogenome data. In the paragraph 2 of the introduction we focus on the differences of reported phylogenetic relationships of Macroheterocera, because it seems that use of various molecular markers and limited data lead to phylogenetic differences in Macroheterocera. Whole mitogenomes contain high phylogenetic information which provide increased resolution at different taxonomic levels. Thus, the full mitogenome has the potential to be used as a unified molecular marker for phylogenetic analysis. Furthermore, although the mitogenomes of about 120 macroheteroceran species or subspecies have been published, there still need more new mitogenomes to be described.
- The methods are unclear, authors should clearly describe how they assembled and annotated the mitogenomes. (parameters of mitoz should be stated in the methods section).
Response: We have added a detailed description of the process of assembly and annotation of mitogenomes in the section of Mitogenome sequencing, assembly, and annotation. Also, we have stated the parameters of Mitoz.
- "We analyzed the phylogenetic relationships of the 6 superfamilies in order Lepidoptera, with two Trichoptera species as outgroup."There needs to be a list with metadata for each specimen used in this study including, voucher, sampling coordinates, and species name. For example, it would be interesting which caddisfly mitogenomes were used.
- I could not access the figures and thus did not review these.
Response: Thank you for pointing out this. While we listed all the specimen information you suggested in Table 1 in the manuscript. We are not sure if all the tables and figures were not accessible and we would work out this problem in time. We are sorry for the inconvenience caused to you.
- Raw reads as well as mitogenomes need to be uploaded to NCBI SRA or a similar data portal, otherwise, the study is not reproducible and accession numbers need to be given in a table or appendix.
Response: The complete 17 mitogenomes were submitted to GenBank, and the accession numbers in bold beginning with OP were shown in Table 1.
Yours faithfully,
Chuang Zhou

Reviewer 2 Report
Manuscript addresses topic of common interest, is original and contains novel data and analysis. Phylogenetic relationships within lepidoptera still known insufficiently in spite of economic and social significance of the topic.
Still I have several concerns about the MS:
General remarks
Phylogenetic inferences:
1. The choice of partition scheme and model(s) of molecular evolution are not given is sufficient details. In order to estimate the correctness of the inference it is important to know if the same model was applied uniformly or individually to each partition etc., were there significant differences in the evolutionary rates between partitions/genes? In general, this part needs more details.
2. Often basal branching order depends strongly on the OTU choice. Therefore the results would be more convincing if the authors tested directly the significance of of the branching order of the competeng phylogenetic trees using respective topological constraints with MrBayes of BEAST.
Structural comparison of mitogenomes
Unfortunately the structural reasons for differences in the genomes lengths are not given. Interesting if it is due to differences in intergenic spacers, control region or something else.
Recently the evidences for remolding of mitocondrial tRNA are accumulating, thus it would be interesting to check if trnM-trnI-trnQ to trnI-trnQ-trnM transition happened as a some sort of transposition or as a remolding event, but this is just a suggestion.
Phylogenetic relationships among the major groups
While full mitogenome based phylogenies mostly are very good, there is always a danger to come across a transgression event, as it takes place in case of the Chironomids. I believe that it should be mentioned in Discussion
Editorial:
I have only one suggestion: please use "inference" instead of "reconstruction" of a phylogeny
Author Response
Dear Reviewer,
Thank you for your comments concerning our manuscript entitled “Characterization of seventeen complete mitochondrial genomes: structural features and phylogenetic implications of the Lepidopteran insects ” (insects-1952476) . Those comments are valuable and very helpful. We have read through comments carefully and have made corrections. Based on the instructions provided in your letter, we uploaded the file of the revised manuscript. Revisions in the text are shown using red highlight for additions, and strikethrough font for deletions. The responses to the reviewer's comments are marked in red and presented following.
We would love to thank you for allowing us to resubmit a revised copy of the manuscript and we highly appreciate your time and consideration.
General remarks
Phylogenetic inferences:
- The choice of partition scheme and model(s) of molecular evolution are not given is sufficient details. In order to estimate the correctness of the inference it is important to know if the same model was applied uniformly or individually to each partition etc., were there significant differences in the evolutionary rates between partitions/genes? In general, this part needs more details.
Response: Thank you for pointing this. We add more details into the manuscript as your suggestions. And the revised statements are as follows:
The best-fit partitioning strategy and models were selected using the greedy search by Par-titionFinder [30]. As a result, twelve subsets partition schemes for the PCG data matrix were obtained for BI analyses (subset1-8,10,12: GTR+I+R, atp6-8, cox1-3, nad1, nad2, nad4 and nad6; subset9,11: TIM+I+G, nad4l and nad5). The MrBayes v3.2.7 [31] was used to phylogenetic inference using BI methods with four independent Markov chains run for 10,000,000 generations and below an average standard deviation value of 0.01. The IQ-TREE [32] in Phylosuite was employed to phylogenetic inference using ML methods with automatic model prediction and 1,000 ultrafast bootstraps.
- Often basal branching order depends strongly on the OTU choice. Therefore the results would be more convincing if the authors tested directly the significance of the branching order of the competeng phylogenetic trees using respective topological constraints with MrBayes of BEAST.
Response: Thank you for your suggestion. We are sorry for that we couldn’t finished the phylogenetic trees with MrBayes of BEAST during limited time. We finished the phylogenetic trees by MrBayes v3.2.7 though the partition strategy and models were selected by PartitionFinder in Phylosuite. Similar method was extensively adopted in insect mitogenome phylogenetic constructions (Miao 2020; Li 2020; Zheng 2022). Thus, the phylogenetic result was probably reliable.
Miao, Xiaoqian, Junhao Huang, Frank Menzel, Qingyun Wang, Qiaoyu Wei, Xiao-Long Lin, and Hong Wu. "Five mitochondrial genomes of black fungus gnats (Sciaridae) and their phylogenetic implications." International Journal of Biological Macromolecules 150 (2020): 200-205.
Li, Xin-yu, Li-ping Yan, Thomas Pape, Yun-yun Gao, and Dong Zhang. "Evolutionary insights into bot flies (Insecta: Diptera: Oestridae) from comparative analysis of the mitochondrial genomes." International Journal of Biological Macromolecules 149 (2020): 371-380.
Zheng, Xiaofeng, Rusong Zhang, Bisong Yue, Yongjie Wu, Nan Yang, and Chuang Zhou. "Enhanced Resolution of Evolution and Phylogeny of the Moths Inferred from Nineteen Mitochondrial Genomes." Genes 13, no. 9 (2022): 1634.
- Structural comparison of mitogenomes
Unfortunately the structural reasons for differences in the genomes lengths are not given. Interesting if it is due to differences in intergenic spacers, control region or something else.
Response: Thank you for your suggestions to make the manuscript better. We have added an analysis of the causes of structural differences in mitogenomes in the first paragraph of the Discussion.
- Recently the evidences for remolding of mitocondrial tRNA are accumulating, thus it would be interesting to check if trnM-trnI-trnQ to trnI-trnQ-trnM transition happened as a some sort of transposition or as a remolding event, but this is just a suggestion.
Response: This research question is really interesting. In particular, the latest research found for the first time in chironomids that mitochondrial gene rearranged from trnI-trnQ-trnM to trnI-trnM-trnQ. We have added this research in the first paragraph of the Discussion. For the study of remolding of mitochondrial tRNA, we believe that we should combine the mitochondrial genomes of different groups of insects for comprehensive analysis, but our mitochondrial genome data are limited. Anyway, this may be our future research direction.
- Phylogenetic relationships among the major groups
While full mitogenome based phylogenies mostly are very good, there is always a danger to come across a transgression event, as it takes place in case of the Chironomids. I believe that it should be mentioned in Discussion
Response: Thank you for your suggestions to make the manuscript better. In the last part of the discussion: 4.3 Phylogenetic relationships among the major Macroheterocera groups, we mentioned the poor signal for phylogenetic reconstruction at the subfamily level in the chironomids.
I have only one suggestion: please use "inference" instead of "reconstruction" of a phylogeny
Response: Thank you for the good suggestion. We have revised the text in the revised manuscript.
Thanks again to the reviewers and editors for the time spent on our article!
Yours faithfully,
Chuang Zhou

Round 2
Reviewer 1 Report
While the authors stated the accession numbers of the mitogenomes, I could not find the accession number of the raw data uploaded to the sequence reads archive of NCBI or elsewhere. The authors have to provide the raw sequencing data
Best,
J.
Author Response
Thanks for your suggestion. We have uploaded the raw sequence data of seventeen insects to NCBI (https://www.ncbi.nlm.nih.gov/), which is available in this project PRJNA892246 for release upon publication.
Reviewer 2 Report
I find it possible to recomment the ms for publication in it's present form
Author Response
Thanks very much for your kind suggestions and comments to improve our paper. On behalf of my co-authors, we highly appreciate your time and consideration.